# Metabolome and Transcriptome Integrated Analysis of *Mulberry* Leaves for Insight into the Formation of Bitter Taste

**DOI:** 10.3390/genes14061282

**Published:** 2023-06-17

**Authors:** Jin Huang, Yong Li, Cui Yu, Rongli Mo, Zhixian Zhu, Zhaoxia Dong, Xingming Hu, Wen Deng

**Affiliations:** Cash Crops Research Institute, Hubei Academy of Agricultural Sciences, Wuhan 430064, China; huang_jin@hbaas.com (J.H.); liyong8057@hbaas.com (Y.L.); yucui_b2020@163.com (C.Y.); monamorus@hbaas.com (R.M.); zhuzhixian@hbaas.com (Z.Z.); dongzhaoxia9@hbaas.com (Z.D.); 13607121598@163.com (X.H.)

**Keywords:** mulberry leaves, bitter taste, sugar, metabolome, transcriptome

## Abstract

*Mulberry* leaves are excellent for health care, confirmed as a ‘drug homologous food’ by the Ministry of Health, China. The bitter taste of mulberry leaves is one of the main problems that hinders the development of the mulberry food industry. The bitter, unique taste of mulberry leaves is difficult to eliminate by post-processing. In this study, the bitter metabolites in mulberry leaves were identified as flavonoids, phenolic acids, alkaloids, coumarins and L-amino acids by a combined analysis of the metabolome and transcriptome of mulberry leaves. The analysis of the differential metabolites showed that the bitter metabolites were diverse and the sugar metabolites were down-regulated, indicating that the bitter taste of mulberry leaves was a comprehensive reflection of various bitter-related metabolites. Multi-omics analysis showed that the main metabolic pathway related to bitter taste in mulberry leaves was galactose metabolism, indicating that soluble sugar was one of the main factors of bitter taste difference in mulberry leaves. Bitter metabolites play a great role in the medicinal and functional food of mulberry leaves, but the saccharides in mulberry leaves have a great influence on the bitter taste of mulberry. Therefore, we propose to retain bitter metabolites with drug activity in mulberry leaves and increase the content of sugars to improve the bitter taste of mulberry leaves as strategies for mulberry leaf food processing and mulberry breeding for vegetable use.

## 1. Introduction

*Mulberry* (*Morus* spp.) is an important economic crop widely distributed from temperate to subtropical regions [1]. *Mulberry* leaves are rich in a variety of nutrients and bioactive substances, including sugar, protein, fiber, vitamins, minerals, polyphenols, alkaloids and amino acids required for the human body [2,3]. For thousands of years, mulberry leaves have been used as feed for silkworms and in traditional Chinese medicine [4,5]. However, mulberry leaves are excellent for health care, confirmed as a ‘drug homologous food’ by the Ministry of Health, China [6]. They have a high edible nutritional value as well as antibacterial, anti-tumor, anti-aging, blood sugar and blood pressure lowering effects, and other medicinal effects [5,7,8,9]. Therefore, mulberry leaves are an ideal source of functional food. In recent years, mulberry leaves have been widely used as raw-food materials, such as mulberry leaf tofu, mulberry leaf ice cream, mulberry leaf noodles, mulberry leaf biscuits and mulberry leaf drinks made with mulberry leaf powder as one of the raw materials, and mulberry bud tea made of mulberry buds and leaves as raw materials [10,11,12]. Moreover, mulberry buds and young leaves can also be directly made into mulberry sprout dishes as vegetables to eat. Various kinds of mulberry sprout dishes have gradually emerged in southern China, appearing in restaurants and on people’s tables.

The bitter taste of mulberry leaves is a problem when used as food. Bitterness is the least popular of the five tastes, and mammals generally dislike or even reject bitter foods [13,14]. The bitter taste of mulberry leaves also limits their use as fodder. In plants, bitter substances are ubiquitous, including polyphenols, alkaloids, amino acids and peptides, saponins and inorganic salts [15,16,17]. Different plants exhibit different levels of bitterness depending on the type and content of the bitter substances they possess [18]. There may be taste interactions among different tastes and among different bitter compounds, which makes it difficult to study the formation factors of bitter taste in mulberry leaves. Different tastes will produce different effects when mixed. Studies have shown that both umami and sweet taste have a masking effect on bitterness [19,20]. For example, the mellow bitterness of chocolate is characteristic of its quality, and to maintain this bitterness during processing, appropriate amounts of sugar can be added to equalize the bitterness without removing it [21]. Therefore, from a single bitter substance, it is difficult to determine the formation of bitter mulberry leaves. The bitter taste of mulberry leaves may be a comprehensive reflection of various metabolites.

The rise of metabolomics provides a new research strategy for the formation of bitter taste in mulberry leaves. MMHub, a database for the mulberry metabolome, annotates 124 metabolites of mulberry leaves, among which 90 metabolites contain detailed information about related chemical structures, including alkaloids, flavonoids, bitter amino acids, etc., related to the bitter taste of mulberry leaves [22]. Metabolomics contains more comprehensive information and is more conducive to the analysis of complex bitter-related metabolites in mulberry leaves [22]. However, the types and quantities of metabolites related to bitterness are very large, which makes it difficult to analyze the types or compounds of the key metabolites that determine the bitterness of mulberry leaves. The use of RNA-Seq to predict and verify new key regulators, and direct and indirect targets in plant metabolite synthesis pathways, has become a new comprehensive information analysis method [23,24,25]. Combined metabolome and transcriptome analysis is advantageous in identifying key metabolites, metabolite synthesis pathways and new key regulatory factors. For example, combined transcriptome and metabolome analysis showed that the flavonoid biosynthesis pathway and isoflavone biosynthesis pathway were the most significant enrichment pathways in the infection response of thrips [26]. Furthermore, the key structural genes and transcription factors regulating the metabolism of soluble sugars, organic acids and important volatiles in kiwifruit were identified via a comprehensive analysis of the metabolome and the whole genome transcriptome, and the regulatory mechanisms of the key transcription factors regulating flavor metabolism were also revealed [27]. The combined analysis method of the transcriptome and metabolome has a good application in the flavonoid biosynthesis of mulberry. Li et al. analyzed the distribution of flavonols and flavonoids in 91 mulberry resources, and identified the key enzyme of rutin synthesis, flavonol 3-O-glucoside-O-rhamnosyltransferases (FGRTs), combined with metabolic spectrum analysis [28]. Xu et al. conducted a joint analysis of the metabolome and transcriptome of mulberry leaves harvested at different times, and found that flavonoids were significantly enriched after frost, and that the expression level of unigenes related to flavonoid biosynthesis was also increased, suggesting that low temperature may be the key triggering factor for the flavonoid biosynthesis of mulberry leaves by improving the expression of genes related to flavonoid biosynthesis [29].

In this study, metabolites related to bitterness in mulberry leaves were analyzed by comparing the metabolites of leaves with different bitterness. The genes and pathways involved in the synthesis of bitter substances in mulberry leaves were analyzed via comparative transcriptome. Combined with metabolome and transcription analysis, the main metabolites, genes and their synthetic pathways affecting the bitter taste of mulberry leaves were analyzed, and the physiological mechanism of the formation of the bitter astringency of mulberry leaves was preliminarily determined. This provided a theoretical basis for mulberry breeding for food mulberry leaves.

## 2. Materials and Methods

### 2.1. Plant Materials

*Mulberry* (*Morus* spp.) plants were grown at the Central China Branch of National *Mulberry* Germplasm Resources, located in Wuhan, Hubei Province, China. A total of 18 *Mulberry* resources were tested for sensory evaluation (Appendix A). Three trees of each resource were sampled. The second leaf from the terminal bud was picked for each plant, and there was 1 replicate for each plant, with 3 replicates in total. The young leaves were well grown and free from pests and diseases. The leaves of the mulberry resources *Morus alba* L. var. *B436* (B436) and *M. alba* L. var. *heyebai* (HYB) were collected from the mulberry plants for metabolome and transcriptome analysis.

### 2.2. Bitter Taste Levels Evaluation

Young leaves from different varieties of mulberry were picked, cleaned, and boiled for 1 min without any seasoning. The bitterness level of mulberry leaves was then rated from 1 to 9. We defined significant differences in the bitterness levels of the white bud of Guisang12, Taiwan No. 2 and 3 mulberry resources as 2, 4 and 8. The young leaves of three mulberry resources were presented to a trained sensory panel (4 males and 3 females) in sequence after being treated as described above, and the bitterness level of each mulberry leaf was judged using a triangulation test. The group members who completed mulberry leaf bitterness sensory training rinsed their mouths with purified water before and after sample evaluation. The sample evaluation interval was 5 min, and each sample was evaluated twice. The significance of the bitterness level of mulberry leaves was analyzed using SPSS 22.0 software.

### 2.3. UPLC–MS/MS System and Analytical Conditions

The sample extraction and professional analytical processes of metabolite detection were performed at Metware Biotechnology Co., Ltd. (Wuhan, China) following their standard procedures [30].

### 2.4. Metabolomics Data Analysis

The qualitative and quantitative mass spectrum data were analyzed using Analyst 1.6.3 software (AB SCIEX, Concord, ON, Canada). Unsupervised principal component analysis (PCA) was performed within the statistical function prcomp in R software (www.rproject.org, accessed on 15 September 2021), and the data were scaled via unit variance before PCA analysis. The cor function and ComplexHeatmap packages of R software were used to calculate the Pearson correlation coefficient (PCC) between samples and the hierarchical cluster analysis (HCA) of samples and metabolites, respectively. R-package MetaboAnalystR was used to generate VIP values of OPLS-DA results, which also included score plots and permutation plots. Significantly regulated metabolites between groups were determined using VIP ≥ 1 and |log_2_ fold change| ≥ 1. The identified metabolites were annotated using the KEGG compound database, and then mapped to the KEGG pathway database. The pathways with significantly regulated metabolites were then input into MSEA (metabolite set enrichment analysis), and their significance was determined using the *p*-values of hypergeometric tests.

### 2.5. Transcriptome Data Analysis

Libraries were constructed from enriched mulberry leaf mRNA and sequenced using the Illumina HiSeqTM 4000 platform (Illumina, San Diego, CA, USA) by Metware Biotechnology Co., Ltd. (Wuhan, China). The *Mulberry* reference genome and its annotation file were downloaded from the NCBI website, and HISAT2 software (v2.2.1) was introduced to map reads to the reference genome of *Morus notabilis* (taxid:981085). The mapped reads of each sample were assembled using StringTie (v1.3.4d) with default parameters and gffcompare software (v2.2.1) to reconstruct a comprehensive transcriptome from all transcriptomes of each sample. The novel genes were predicted by assembling the mapped reads and comparing them with the original genome. To obtain the annotation information, the unigenes were compared with public databases, including KEGG, GO, KOG, NR, Swiss-Prot, Pfam and Trembl. To evaluate gene expression levels, transcript abundances were calculated and normalized to FPKM (fragments per kilobase of transcript per million fragments mapped). featureCounts v1.6.2 was used to calculate the FPKM of each gene based on the gene length. PCA was performed within the statistical function prcomp in R software to obtain the relationships among, and variability between, samples. The differential expressions between the two groups were analyzed using DESeq2 (v1.22.1), and the *p* value was corrected using the Benjamini and Hochberg method. The corrected *p* value < 0.05 and |log2 fold change| ≥ 1 were used as the threshold for significant difference expression. Enrichment analysis was conducted based on a hypergeometric test. For KEGG, a hypergeometric distribution test was conducted with the pathway as a unit. For GO, it was based on the GO term.

### 2.6. Integration Analysis of Metabolome and Transcriptome

We performed integration analyses of differentially accumulated metabolites (DAMS) and differentially expressed genes (DEGs) from metabolic fractions and transcriptomes to determine the enrichment of pathways. The transcription–metabolite network was constructed using the gene metabolite network with a Pearson correlation coefficient (PCC) > 0.8.

### 2.7. Quantification of Soluble Sugars in Mulberry Leaves

To detect the concentration of soluble sugars, 100 mg leaves were sampled and ground into a homogenate with 1 mL distilled water. After being incubated at 80 °C for 20 min, they were centrifuged at 12,000× *g* for 20 min and the supernatant was collected. All supernatants were diluted 20 times for the determination of soluble sugars. In total, 40 µL of diluted sample solution was added to 40 µL distilled water, 20 µL anthrone-sulfuric acid solution, and 200 µL concentrated sulfuric acid, mixed, and then placed in a 95 °C water bath for 10 min. After cooling to room temperature, 200 µL was taken from a 96-well plate at 620 nm to determine its absorbance value. The 5 mg/mL glucose standard solution was diluted into 0.8, 0.6, 0.5, 0.4, 0.3, 0.2 and 0.1 mg/mL gradient solutions, which were measured at the same time as the sample, and the standard curve was drawn according to the relationship between the concentration and light absorption value. Total soluble sugar (mg/g) = C × V ÷ M × F. C; total soluble sugar content calculated from the standard curve, mg/mL; V, extract liquid volume, 1 mL; M, sample weight, g; F, dilution ratio. The software GraphPad Prism 5 was used to analyze the Pearson correlation coefficients of the soluble sugars and bitter taste levels.

## 3. Results

### 3.1. Evaluation of Bitter Taste Levels for Different Mulberry Leaves

To evaluate the bitter taste levels of 30 mulberry resources, we set the young leaves of 3 mulberry varieties with increasing bitter taste levels to be 2, 4, and 8 (Figure 1). Based on this standard, the bitter taste levels of the young leaves of the other 27 mulberry resources were evaluated using the multi-person taste evaluation method. The evaluation result is shown in Appendix A. The bitter taste levels were graded from slightly bitter to extremely bitter with an increasing score in the range of 1–9. The bitter taste phenotypes of 30 mulberry varieties basically covered all bitter taste levels, and there was a great difference between the mulberry varieties with the highest bitter taste level and the lowest. SPSS 22.0 software was used to analyze the significance of the bitter taste levels of mulberry leaves of various varieties (*p* ≤ 0.05). The mulberry variety with the highest bitter taste out of all the mulberry leaves was *Heyebai*, followed by *Dabaiya* and *Ribentiancheng*. The two varieties with the lowest bitter taste of mulberry leaves were *Bai436* and *Bai185*. The mildly bitter mulberry variety *Bai436* (B436) and the extremely bitter mulberry variety *Heyebai* (HYB) were selected for wide-target metabolomics and transcriptomics analysis (Figure 1).

### 3.2. Transcriptome and Metabolome Analysis of Mulberry Leaves

The second tender leaves from the top of the annual branches of B436 and HYB were taken for broad target metabolomics and transcriptomics analysis. Three independent biological replicates were used for each variety, resulting in six samples. In transcriptome detection, a total of 40.99 Gb of Clean Data was obtained, and the Clean Data of each sample were above 6.53 Gb, and the percentage of the Q30 base value was over 94.05% (Table 1). Using HISAT v2.1.0, clean reads were compared to the reference genome with an efficiency of over 71.46%. A total of 20,761 transcripts were obtained. According to the location information of genomic reads, the reads were assembled into 3792 new transcripts using StringTie v1.3.4d. All transcripts obtained from the transcriptome were compared with seven major databases (Nr, Swiss-prot, Pfam, KOG, GO, Tremble and KEGG databases). A total of 22,567 homologous genes were identified in NR (16,546), Swiss-prot (16,871), Pfam (19,086), KOG (12,956), GO (18,953), Tremble (13,980), and KEGG (16,372) data from 24,553 transcripts’ homologous genes were found in the library (Appendix A). In addition, 872 metabolites were detected, which can be divided into 11 broad categories (Appendix A).

The principal component analysis (PCA) of DEGs and DAMs showed significant differences between the B436 and HYB groups, explaining 51.3% and 71.8% of the total variation, respectively (Figure 2A,B). These results indicated that there were significant differences in metabolites and genes related to the bitter taste of mulberry leaves between B436 and HYB.

### 3.3. DEGs and DAMs Related to Bitter Taste in Mulberry Leaves

DESeq2 v1.22.1 was used to analyze the DEGs between the two groups. B436 compared with the HYB of mulberry leaves was detected in 3809 DEGs, of which there were 2231 and 1578 DEGs raised and lowered (B436 vs. HYB, *p*-adjusted value < 0.05 and |log2FC| ≥ 1, Figure 3A). We used an OPLS-DA model to evaluate the differential accumulative metabolites between B436 and HYB mulberry young leaves. The established OPLS-DA model had a good fit (Appendix A). A total of 296 differential metabolites were detected between the two groups, with 181 metabolites up-regulated and 115 metabolites down-regulated (B436 vs. HYB, VIP ≥ 1 and |log2FC| ≥ 1, Figure 3B). Further analysis revealed 2892 gene annotations on 131 different KEGG pathways (Appendix A). The enrichment of the KEGG pathway showed that the differential genes related to the bitter taste of mulberry leaves were mainly involved in plant pathogen interactions and the MAPK signaling pathway (corrected *p*-value < 0.05, Figure 4A).

For DAMs, we found that DAMs were most significantly enriched in the linoleic acid metabolic pathway, pyrimidine metabolic pathway, galactose metabolic pathway, α-linoleic acid metabolic pathway, and bioanabolic metabolic pathway of flavonoids and flavonols (Figure 4B). Bitter substances in plants are mainly divided into five categories: polyphenols, alkaloids, amino acids and peptides, saponins and inorganic salts. Furthermore, the results showed that the top 10 up-regulated metabolites in the mulberry young leaves of B436 versus HYB were N-acetyl-L-glutamic acid, 4,4-dihydroxy-3,5-dimethoxybibenzyl acid, cinnamic acid, phenoxyacetic acid, isovanillin, homogentian acid, sacherol 5,3-di-O-glucoside, 4-hydroxy-chalcone, isorhamnose-3-O-gallate and homoplantine-7-O-(6-O-Coumaryl)-glucoside (Figure 5A). Humans have the same receptors for bitter taste and sweet taste and the two taste sensations have a competitive inhibition relationship. The different metabolites of sweet taste between B436 and HYB were mainly sugar alcohols and the 10 sugar alcohol metabolites that were most regulated were 1, 6-dehydrate-β-D-glucose, D-glucose, D-fructose, L-xylose, D-(+)-galactose, inositol, D-arabinose, mannose, isomalmaltose and lactise (Figure 5B).

### 3.4. The Transcriptome and Metabolome Were Combined to Analyze the Metabolic Pathways Related to Bitter Taste in Mulberry Leaves

To quantitatively analyze the transcripts of metabolic pathways associated with the bitter taste of mulberry leaves, we performed a combined transcriptome and metabolome KEGG pathway enrichment analysis. The results showed that DEGs and DAMs were simultaneously enriched in the galactose metabolic pathway (*p* ≤ 0.05, Figure 6). To better understand the relationship between genes and metabolites, we mapped both DEGs and DAMs to the KEGG pathway map (Appendix A). As shown in Appendix A, 35 key genes (19 up-regulated and 16 down-regulated) and 9 metabolites (2 up-regulated and 7 down-regulated) were simultaneously mapped to the galactose metabolic pathway (ko00052). We selected certain key DAMs and DEGs from the galactose metabolic pathway to remap the gene regulatory network (Figure 7). From the expression level, most of the key pathway genes involved in the synthesis of D-Dalactose and D-Glucose were significantly changed. Indeed, GOLS (three DEGs), Raffinose synthase (two DEGs), Stachyose synthetase (one DEG), INV (three DEGs), GLA (one DEG) and lacZ (one DEG) encoding enzymes were more highly expressed in B436 than in HYB, denoting that these DEGs are the key genes involved in the synthesis of D-Dalactose and D-Glucose. On the contrary, the key pathways of one GOLS gene, two Raffinose synthase genes, seven INV genes, two GLA genes, one MGAM gene and six lacZ genes were significantly up-regulated, implying that silencing these genes contributes to the accumulation of D-Dalactose and D-Glucose in mulberry leaves.

### 3.5. Soluble Sugar Is One of the Main Factors Responsible for the Difference in the Bitter Taste of Mulberry Young Leaves

According to the multi-omics analysis, the content of sweet-taste metabolites may be closely related to the difference in bitter taste between the two mulberry young leaves. We detected the soluble sugar content in the young leaves of six mulberry resources with different bitterness levels, as shown in Figure 8A. The soluble sugar content was highest in B436 mulberry leaves, but lowest in HYB in six mulberry resources. In order to analyze the relationship between soluble sugar and the bitter taste of mulberry leaves, we performed a correlation analysis between the soluble sugar and bitter taste level. The soluble sugar content in mulberry leaves was strongly correlated with the bitter intensity of mulberry leaves (R = −0.77, *p* = 0.0002), indicating that the soluble sugar content was one of the main factors for the bitter astringency difference in mulberry leaves (Figure 8B).

## 4. Discussion

Bitter taste is one of the basic tastes, and its most prominent feature is that it is easily perceived by people with a very low threshold [31]. Among woody plants, tea tree has the most studies on bitterness. Bitter substances in tea include tea polyphenols (catechin, epicatechin, epicatechin, epicatechin gallate, epicatechin gallate, epicatechin gallate), tannins, phenolic acids, flavonoids, alkaloids, amino acids, etc. Catechins are the main bitter substances [32,33,34]. Based on the transcriptome and metabolome analysis of mulberry leaves B436 and HYB, the DEGs and DAMs were studied to explore the formation mechanism of bitter taste in mulberry leaves. Our results indicate that the bitter substances in mulberry leaves are flavonoids, phenolic acids, alkaloids, amino acids and so on. Compared with tea leaves, tea polyphenols were not detected in mulberry leaves, and tannins were less detected. Only ellagic acid 4-O-glucoside was significantly different between the two kinds of mulberry leaves, indicating that the bitter formation mechanism of mulberry leaves was very different from that of tea leaves.

Bitter substances in mulberry leaves are very varied. In this study, the changes in metabolites related to bitter taste in the DAMs of the young mulberry leaves of two mulberry varieties were very complex. Compared with HYB, 41 kinds of phenolic acids were up-regulated and 21 were down-regulated in B436. There were nine kinds of L-type amino acids, of which six were up-regulated and three were down-regulated. There were 68 flavonoids, 38 of which were up-regulated and 29 which were down-regulated. There were 16 kinds of alkaloids, 8 of which were up-regulated and 8 of which were down-regulated. These results indicate that the formation mechanism of bitter taste in mulberry leaves is the comprehensive embodiment of a variety of bitter metabolites [35]. The L-type amino acids (six up-regulated and three down-regulated), coumarin (four up-regulated and one down-regulated), kaferol glycoside (seven up-regulated and two down-regulated) in DAMs were consistent with the bitter astringency phenotype of mulberry leaves, and were speculated to play an important role in the formation of bitter taste in mulberry leaves [36].

In the food industry, adding an appropriate amount of sweetness to bitterness can suppress part of the bitterness and make the overall taste mild and delicate [37,38]. A total of 26 kinds of sugar alcohol metabolites were detected in this study, of which 16 species were significantly different in the leaves of the 2 mulberry varieties, among which 3 kinds were up-regulated and 13 kinds were down-regulated. In this study, 35 glucose metabolizer genes (19 up-regulated and 16 down-regulated) and nine glucose metabolites (two up-regulated and seven down-regulated) were also located in the galactose metabolic pathway in the joint transcriptome and metabolome analysis. These results indicated that carbohydrates had a significant inhibitory effect on the bitterness of mulberry leaves.

To sum up, bitter substances in mulberry leaves are diverse and complex, and the formation of a bitter taste in mulberry leaves is a comprehensive reflection of a variety of bitter metabolites. In this study, many amino acids, flavonoids, phenolic acids, alkaloids and other metabolites related to the bitter taste of mulberry leaves detected were found to have drug activity, and have medicinal effects such as antibacterial, anti-tumor and anti-aging effects, as well as blood sugar and blood pressure lowering effects [39,40,41]. As a functional food, and also the origin of both a medicine and food, these bitter compounds play a significant role in mulberry leaves. Traditional Chinese medicine believes that bitter compounds can relieve asthma, stop vomiting, promote defecation and have other effects [42,43]. At the same time, bitter taste is also indispensable in seasoning, as it can enrich or even improve the flavor of food when matched with other tastes [13,44]. Sweet and bitter tastes have a competitive inhibition relationship [19,20,37]. In this study, sugar alcohol metabolites with a sweet taste also showed great differences between the two mulberry leaves, indicating that sugars in mulberry leaves have a great influence on the formation of the bitter taste of mulberry. Therefore, we proposed to preserve bitter metabolites with drug activity in mulberry leaves and increase the content of sugars to improve the bitter aftertaste of mulberry leaves as a strategy for the food processing of mulberry leaves and breeding of mulberry trees for vegetable use.

## Figures and Tables

**Figure 1 genes-14-01282-f001:**
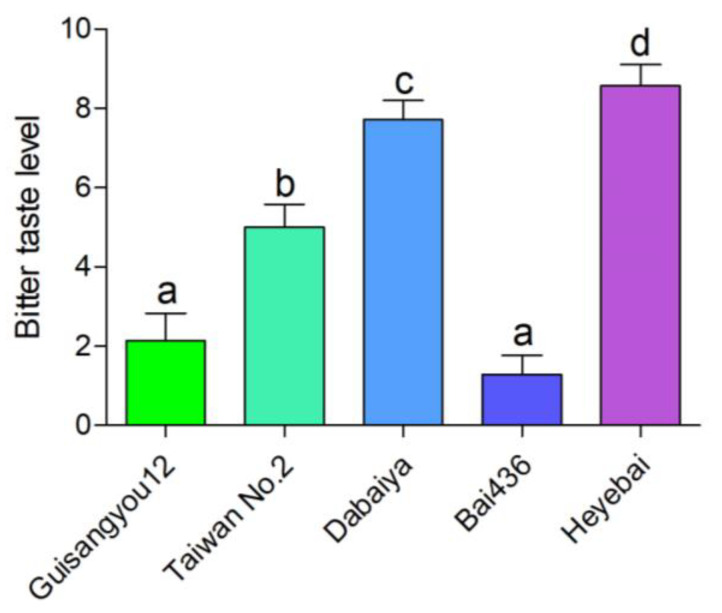
Evaluation of bitter taste levels for different mulberry young leaves. Data represent means ± SD of seven repeats. Different letters above the bars indicate significant differences, as determined using a Tukey honest significant difference test (*p* < 0.05).

**Figure 2 genes-14-01282-f002:**
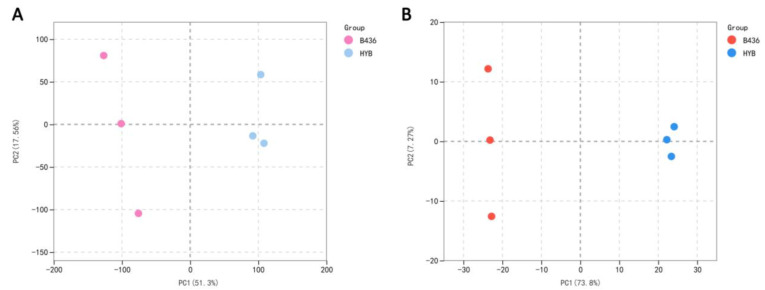
PCA of the variance-stabilized estimated raw counts of DEGs (**A**) and DAMs (**B**) between B436 and HYB.

**Figure 3 genes-14-01282-f003:**
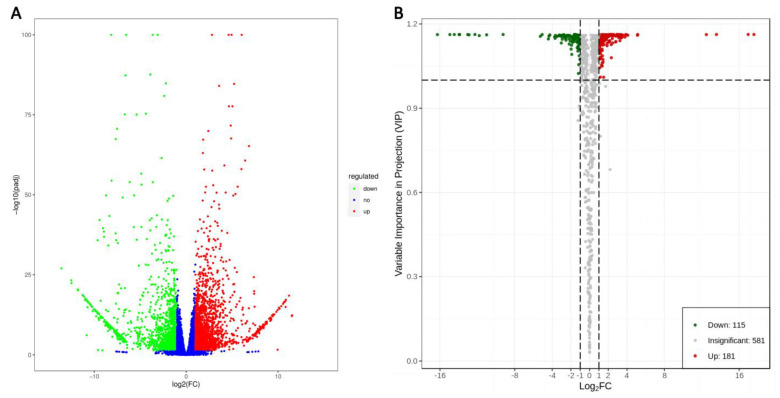
Volcano map of DEGs and DAMs. (**A**) Volcano plots were used to display the down-regulated, non-regulated and up-regulated genes for HYB vs. b436. (**B**) Volcano plots were used to display the down-regulated, insignificant and up-regulated metabolites for HYB vs. b436. The red, green and blue/gray dots represent up-regulated DEGs or DAMs (up), down-regulated DEGs or DAMs (down), and unchanged DEGs or DAMs (no/insignificant), respectively.

**Figure 4 genes-14-01282-f004:**
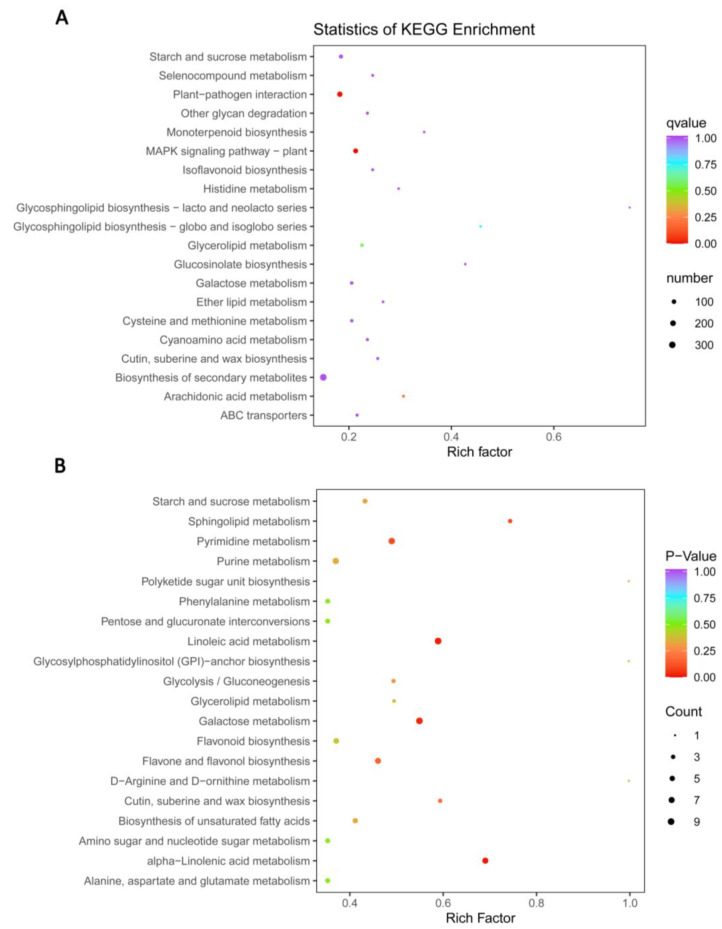
KEGG pathway classification of DEGs and DAMs. (**A**) KEGG pathway classification of DEGs in HYB vs. B436. (**B**) KEGG pathway classification of DAMs in HYB vs. B436.

**Figure 5 genes-14-01282-f005:**
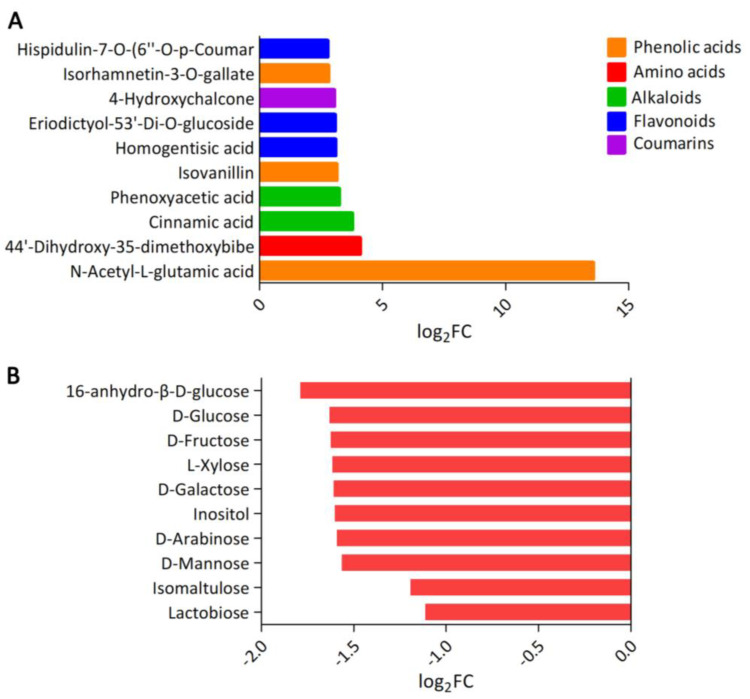
The DAMs analysis in HYB vs. B436. (**A**) Top 10 most accumulated bitter-related metabolites in HYB vs. b436. Different colors represent the classification of metabolites. (**B**) Top 10 least accumulated sugar alcohol metabolites in HYB vs. B436.

**Figure 6 genes-14-01282-f006:**
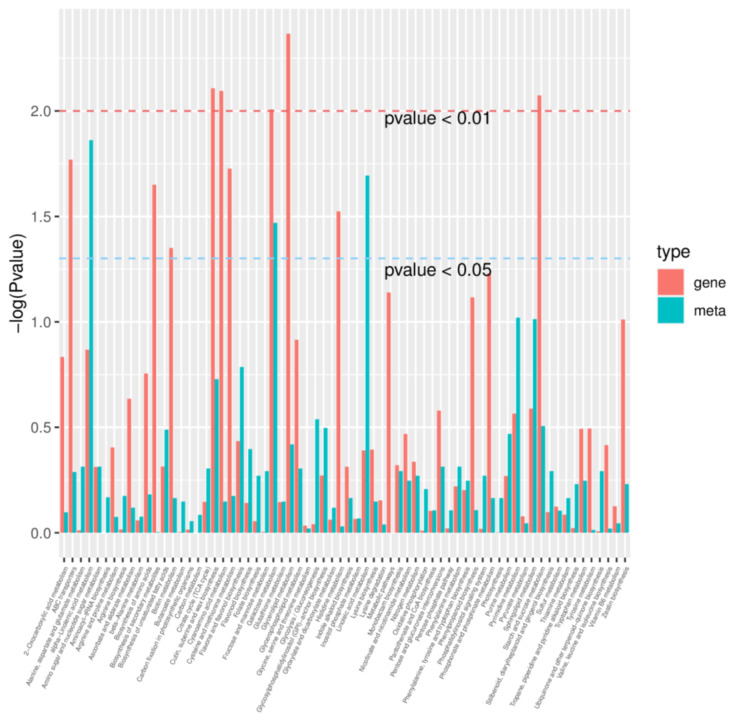
Joint analysis of the DEGs and DAMs between B436 and HYB. Blue line represents the selected gene and metabolic pathways at *p*-value < 0.05, and red line represents the selected gene and metabolic pathways at *p*-value < 0.01.

**Figure 7 genes-14-01282-f007:**
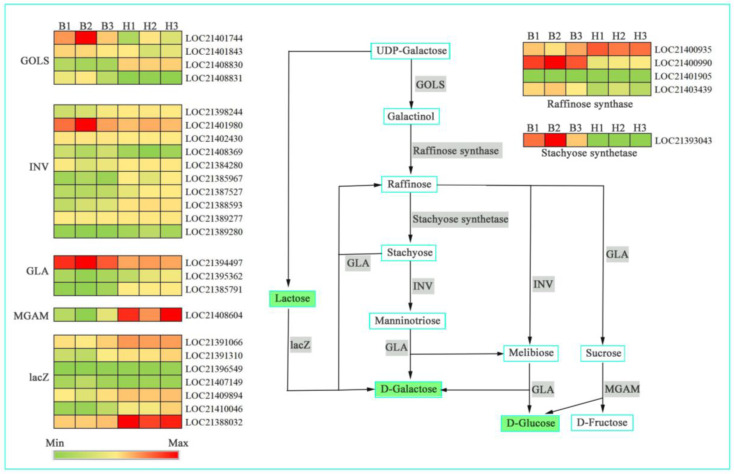
Gene regulatory network related to galactose metabolic pathway. Metabolites and genes are shown in turquoise boxes and grey shades, respectively. Green shades represent down-regulated DAMs. The heat map displays down-regulated and up-regulated structural genes. B1–3, 3 biological replicates of B436; H1–3, 3 biological replicates of HYB. GOLS, galactinol synthase; INV, β-fructofuranosidase; GLA, α-galactosidase; MGAM, α-glucosidase; lacZ, β-galactosidase.

**Figure 8 genes-14-01282-f008:**
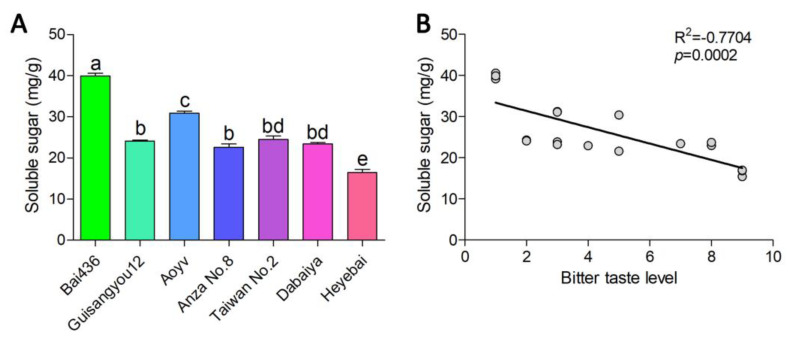
Detection of soluble sugar and its correlation with bitter taste for mulberry young leaves from six mulberry resources. (**A**) Detection of soluble sugar for mulberry young leaves from six mulberry resources. Data represent means ± SD of seven repeats. Different letters above the bars indicate significant differences, as determined by a Tukey honest significant difference test (*p* < 0.05). (**B**) Scatter plots of bitter taste level and soluble sugar. Individual data points are shown with linear regression. R^2^ represents Pearson correlation coefficient.

**Table 1 genes-14-01282-t001:** Transcriptome sequencing data for the leaf samples from B436 and HYB.

Sample	Raw Reads (M)	Clean Reads (M)	Clean Base (M)	Q20 (%)	Q30 (%)
B436-1	46.74	44.48	6.67	98.01	94.5
B436-2	47.34	45.43	6.81	97.93	94.05
B436-3	48.55	46.63	6.99	97.93	94.07
HYB-1	45.44	43.50	6.53	97.93	94.29
HYB-2	47.70	45.77	6.87	97.97	94.39
HYB-3	49.39	47.45	7.12	97.91	94.06
Sum	285.15	273.26	40.99		

## Data Availability

Data is contained within the article and Appendix A.

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
