# Peer review of "Metabolome and Transcriptome Integrated Analysis of Mulberry Leaves for Insight into the Formation of Bitter Taste"

_genes, 2023, doi:10.3390/genes14061282_

Round 1
Reviewer 1 Report
The manuscript entitled: “Metabolome and Transcriptome Integrated Analysis of Mulberry Leaves Provide Insights into the Formation of Bitter Taste” contains some information of potential interest to readers.
The paper is well written and organized. Especially the result and discussion is a clearly explained.
PS: My main suggestion to author in current manuscript, if author either choose from sugar or bitter related biosynthetic key genes and confirmed with gene expression via qRT-PCR and correlate with metabolite levels.
Nil
Reviewer 2 Report
This is very interesting study, but I would like to suggests some improvement in the current study:
1. Author should provide all the details related to assembly which include N50, N95.
2. In Introduction author mentioned "Mulberry have antibacterial, lowering blood sugar, blood pressure, blood pressure, anti-tumor, anti-aging and other medicinal effects". Can you please elaborate metabolites or compounds extracted from Mulberry to treat these diseases.
3. Author should provide gene regulatory network by highlighting key candidates.
There were few grammatical errors, can easily be improved.
